# Distinct clinical and immunological profiles of patients with evidence of SARS-CoV-2 infection in sub-Saharan Africa

Ben Morton [1,2,3,12]✉, Kayla G. Barnes [1,4,5,6,7,12], Catherine Anscombe[1,2], Khuzwayo Jere [1,4,8], Prisca Matambo[1], Jonathan Mandolo[1], Raphael Kamng'ona[1], Comfort Brown[1], James Nyirenda[1], Tamara Phiri [9], Ndaziona P. Banda[8], Charlotte Van Der Veer[1,4], Kwazizira S. Mndolo[9], Kelvin Mponda[9], Jamie Rylance [1,2], Chimota Phiri[9], Jane Mallewa[8], Mulinda Nyirenda [8,9], Grace Katha[9], Paul Kambiya[1], James Jafali[1], Henry C. Mwandumba [1,2,8], Stephen B. Gordon[1,2], Blantyre COVID-19 Consortium*, Jennifer Cornick[1,4,12] & Kondwani C. Jambo [1,2,8,12]✉

Although the COVID-19 pandemic has left no country untouched there has been limited research to understand clinical and immunological responses in African populations. Here we characterise patients hospitalised with suspected (PCR-negative/IgG-positive) or confirmed (PCR-positive) COVID-19, and healthy community controls (PCR-negative/IgG-negative). PCR-positive COVID-19 participants were more likely to receive dexamethasone and a beta-lactam antibiotic, and survive to hospital discharge than PCR-negative/IgG-positive and PCR-negative/IgG-negative participants. PCR-negative/IgG-positive participants exhibited a nasal and systemic cytokine signature analogous to PCR-positive COVID-19 participants, predominated by chemokines and neutrophils and distinct from PCR-negative/IgG-negative participants. PCR-negative/IgG-positive participants had increased propensity for *Staphylococcus aureus* and *Streptococcus pneumoniae* colonisation. PCR-negative/IgG-positive individuals with high COVID-19 clinical suspicion had inflammatory profiles analogous to PCR-confirmed disease and potentially represent a target population for COVID-19 treatment strategies.

[1] Malawi-Liverpool-Wellcome Trust Clinical Research Programme, University of Malawi College of Medicine, Blantyre, Malawi. [2] Department of Clinical Sciences, Liverpool School of Tropical Medicine, Liverpool, UK. [3] Liverpool University Hospitals NHS Foundation Trust, Liverpool, UK. [4] Institute of Infection, Veterinary and Ecological Sciences, University of Liverpool, Liverpool, UK. [5] Harvard School of Public Health, Boston, MA, USA. [6] Broad Institute of MIT and Harvard, Cambridge, MA, USA. [7] University of Glasgow MRC Centre for Virus Research, Glasgow, UK. [8] University of Malawi-College of Medicine, Blantyre, Malawi. [9] Department of Medicine, Queen Elizabeth Central Hospital, Blantyre, Malawi. [12] These authors contributed equally: Ben Morton, Kayla G. Barnes, Jennifer Cornick, Kondwani C. Jambo. *A list of authors and their affiliations appears at the end of the paper. ✉email: ben.morton@lstmed.ac.uk; Kondwani.Jambo@lstmed.ac.uk

The coronavirus disease 2019 (COVID-19) pandemic has, to date, been less severe in sub-Saharan Africa compared to Europe and the Americas[1]. However, clinical diagnosis, triage and treatment decisions for COVID-19 patients is particularly challenging in resource-poor settings. The increased transmissibility of new severe acute respiratory syndrome coronavirus 2 (SARS-CoV-2) variants observed in the second wave of the pandemic has made the need to address these challenges even more urgent[2]. Limited access to advanced life-preserving therapies such as mechanical ventilation, and delayed presentation to hospital, is common[3]. This increases clinical and diagnostic complexity as patients may present when severely unwell, and potentially, without polymerase chain reaction (PCR)-detectable SARS-CoV-2 (ref. [4]).

Cytokine dysregulation has been consistently observed in COVID-19 patients in multiple settings[5,6]. Consequently, cytokines, including interleukin-6 (IL-6) and tumour necrosis factor-α (TNF-α), have shown potential prognostic value to guide decisions on clinical management of COVID-19 (refs. [7,8]), but they lack adequate specificity. For example, IL-6 levels may be 10- to 200-fold higher for patients with the hyperinflammatory phenotype of acute respiratory distress syndrome (ARDS) compared to patients with severe COVID-19 (ref. [9]). Beyond the use of single markers, systems approaches have demonstrated immune signatures distinct to COVID-19 aligned with disease severity[10–12]. They have identified a systemic immune signature showing profoundly altered T cells, selective cytokine/chemokine upregulation[10] and monocyte and neutrophil activation in hospitalised COVID-19 patients[11,12]. Further, inflammatory dysregulated IL-2, IL-6, IL-10 and IL-15 are associated with COVID-19-related mortality[13]. However, there are very few studies that have investigated the respiratory tract, the initial site of infection and disease pathogenesis.

SARS-CoV-2 infection is established when the virus binds to angiotensin-converting enzyme 2 (ACE2)-expressing epithelial cells in the nasal mucosa[14]. The few studies reporting mucosal cellular or cytokine responses in COVID-19 have shown cytokine dysregulation and immune cell disruption in the lower airway, correlating with disease severity[15,16]. However, there is a paucity of information on immune responses against SARS-CoV-2 infection in the upper airway[17]. Minimally invasive sampling techniques of the nasal mucosa are well tolerated, and highly useful in human challenge models of respiratory syncytial virus and *Streptococcus pneumoniae*[18,19]. Given importance of the nose in SARS-CoV-2 infection[20], understanding the host–viral interaction at the nasal mucosa could provide additional insights to understand and potentially modulate COVID-19 prognosis.

Immunological responses against SARS-CoV-2 infection at the nasal mucosa are still poorly understood. Here, we analysed nasal mucosa and peripheral blood samples from a cohort of patients admitted to hospital with suspected and/or confirmed COVID-19. These patients were compared with adult healthy community controls. Immunological parameters were studied using 38-multiplex cytokine assays and flow cytometry, while profiling of respiratory pathogens was done using a 33-multiplex polymerase chain reaction (PCR) respiratory pathogen diagnostic panel. Our study provides insights on the cytokine responses in the nasal mucosa following SARS-CoV-2 infection in severe COVID-19 patients and the potential importance of additional confirmatory antibody tests for diagnosis of SARS-CoV-2 PCR-negative patients with high clinical suspicion of COVID-19.

## Results

**Clinical overview.** Between 21 April 2020 and 25 September 2020, 87 patients (median age 47 years, IQR: 34–62) presenting with severe acute respiratory infection (SARI) were recruited, of whom 60 (69.0%) were male (Table 1). SARS-CoV-2 was confirmed by nucleic acid amplification (NAAT) in 41 participants (classified as "PCR-confirmed COVID-19") (Fig. 1a). The remaining 46 participants with suspected disease were NAAT negative (Fig. 1a). Of these, 25 were IgG positive against spike protein 2 and nucleoprotein on serological testing. Using NAAT and antibody test results, we reclassified the patient groups into PCR-confirmed COVID-19, PCR−/IgG+ SARI

**Table 1 Clinical characteristics.**

|  | Units | PCR+ (n = 41) | PCR−/IgG+ (n = 25) | PCR−/IgG− (n = 21) | P value | Healthy control (n = 24) |
|---|---|---|---|---|---|---|
| Male | n (%) | 30 (73) | 18 (72) | 12 (57) | 0.455[a] | 20 (83) |
| Age | median (IQR) | 50 (42–65) | 34 (25–51) | 41 (37–60) | 0.027[b] | 23 (22–25) |
| Days from symptom onset to hospital admission | median (IQR) | 4 (2–7) | 6 (2–13) | 4 (3–14) | 0.814[b] | NA |
| Days from hospital admission to research sample | Median (IQR) | 4 (3–6) | 3 (1–4) | 1 (1–2) | 0.003[b] | NA |
| HIV seropositive | n (%) | 9 (31[c]) | 9 (45[c]) | 9 (64[c]) | 0.110[a] | 0 (0) |
| TB positive | n (%) | 1 (2) | 0 (0) | 0 (0) | 1.000[a] | 0 (0) |
| Malaria positive | n (%) | 2 (5[c]) | 1 (5[c]) | 0 (0[c]) | 0.157[a] | 0 (0) |
| Cardiac disease | n (%) | 13 (36[c]) | 5 (25[c]) | 3 (14) | 0.204[a] | 0 (0) |
| Pulmonary disease | n (%) | 3 (9[c]) | 0 (0[c]) | 0 (0[c]) | 0.433[a] | 0 (0) |
| Oxygen required at enrolment | n (%) | 21 (54[c]) | 12 (48) | 10 (50[c]) | 0.886[a] | 0 (0) |
| ISARIC 4C clinical severity score | Median (IQR) | 5 (4–8) | 4 (3–7) | 6 (5–8) | 0.075[b] | 0 (0–0) |
| UVA score | Median (IQR) | 2 (0–4) | 2 (1–3) | 4 (2–5) | 0.025[b] | 0 (0–0) |
| Beta-lactam antibiotic administered | n (%) | 32 (78) | 13 (52) | 10 (48) | 0.022[a] | NA |
| Steroids administered | n (%) | 26 (63) | 1 (4) | 0 (0) | <0.001[a] | NA |
| Died in hospital | n (%) | 3 (7[c]) | 5 (20[c]) | 9 (43[c]) | 0.009[a] | NA |
| Hospital length of stay for survivors | Median (IQR) | 8 (6–17) | 6 (4–9) | 6 (3–8) | 0.028[b] | NA |

PCR+ is RT-qPCR confirmed SARS-CoV-2 severe acute respiratory infection (SARI); PCR−/IgG− is patients with SARI who were RT-qPCR negative but IgG positive for SARS-CoV-2; and PCR−/IgG− is patients with SARI who were both RT-qPCR negative and IgG negative for SARS-CoV-2. Healthy controls were ambulant patients with no intercurrent illness who attended an outpatient clinic appointment. ISARIC 4C score calculated using six clinical variables available within dataset as urea and C-reactive protein were not available. *UVA* universal vital assessment score (low-income country validated clinical severity score).
[a]Fisher's exact test.
[b]Kruskall–Wallis test.
[c]Proportion (%) positivity calculated using the denominator for individual variables (unknown status classified as missing data).

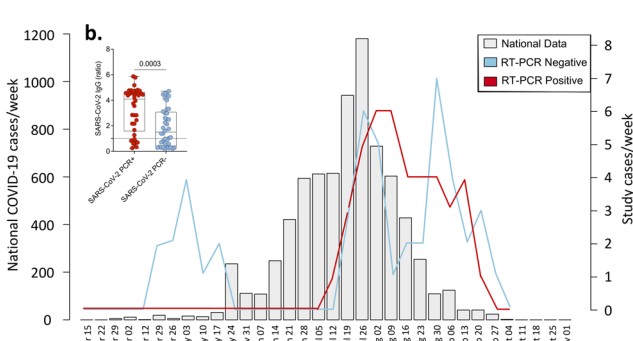

**Fig. 1 Study recruitment timeline and SARS-CoV-2 diagnosis. a** Suspected and SARS-CoV-2 RT-qPCR-confirmed COVID-19 patients recruited at Queen Elizabeth Central Hospital, Blantyre (positive (red) and negative (blue)) reported as cases/week compared to the national data (light grey). National data include SARS-CoV-2 RT-qPCR-confirmed symptomatic and non-symptomatic COVID-19. **b** SARS-CoV-2 Spike protein 2 (S2) and nucleoprotein (NP) IgG antibodies in PCR-negative and -positive individuals. The data are reported as the ratio of OD in the test samples to the assay threshold control. For all boxplots, box boundaries correspond to 25th and 75th percentiles; whiskers extend to a maximum or minimum greatest value. Data were analysed using Mann–Whitney test, two-sided (PCR positive, $n = 38$; PCR negative, $n = 45$). SARS-CoV-2 severe acute respiratory syndrome coronavirus 2, COVID-19 coronavirus disease of 2019, PCR polymerase chain reaction, IgG immunoglobulin, SARI severe acute respiratory infection. Source data are provided as a Source Data file.

($n = 25$) and PCR−/IgG− SARI ($n = 21$) participants (Fig. 1b). In addition, we recruited 25 ambulatory healthy volunteers. Patient demographics and clinical characteristics are reported in Table 1. To verify the specificity of the ELISA assay, we tested pre-COVID-19 pandemic historical sera from individuals with other coronaviruses, malaria convalescent sera, sera from people living with HIV and sera from asymptomatic HIV-uninfected adults, and found a specificity of 8/132 (94% [CI 91–97]) (Supplementary Fig. 1). We also tested antibodies in a WHO-recommended NIBSC Proficiency Plasma Panel (20/120, 20/122, 20/124, 20/126, 20/128 and 20/130) and found 100% concordant results (Supplementary Fig. 1). Hospitalised individuals with PCR-confirmed COVID-19 were significantly older (median 50 [PCR+] vs. 34 [PCR−/IgG+] vs. 41 [PCR−/IgG−], $p = 0.017$), more likely to have received dexamethasone (63% vs. 4% vs. 0%, $p < 0.001$) and more likely to have received a beta-lactam antibiotic (78% vs. 52% vs.48%, $p = 0.022$) compared to the PCR−/IgG+ and PCR−/IgG− SARI participants. Additionally, PCR-confirmed COVID-19 participants were more likely to survive (93% vs. 80% vs. 57%, $p = 0.004$) with increased hospital length of stay (median days 8 vs. 6 vs. 6, $p = 0.028$) compared to the PCR−/IgG+ and PCR−/IgG− SARI participants. PCR−/IgG+ SARI participants also tended toward longer times from symptom onset to hospital admission compared to PCR-confirmed COVID-19 participants (median 6 vs. 4 days) (Table 1).

**Distinct cytokine responses induced by SARS-CoV-2 infection in nasal mucosa and systemic circulation.** We investigated the cytokine response in nasal lining fluid and serum of our four participant groups. Our analysis was limited to 25 PCR-confirmed COVID-19 participants (who were IgG positive), 16 PCR−/IgG+ SARI participants, 11 PCR-/IgG- SARI participants and 25 healthy controls, from whom we had paired nasal lining fluid and serum samples. We measured the concentration

of 37 cytokines (and sCD40L) in paired nasal lining fluid and serum samples of the four study groups (Supplementary Fig. 2 and Supplementary Fig. 3). We found altered cytokine levels in both nasal lining fluid and serum of PCR-confirmed COVID-19 participants, PCR−/IgG+ SARI participants and PCR−/IgG− SARI participants, relative to healthy controls (Fig. 2). Specifically, high concentrations of inflammation-related cytokines were common among all the patient groups compared to healthy controls, including IL-1α, IL-1β, IL-6, MIP-1α and TNF-α in nasal lining fluid, and raised levels of IL-6, IL-10, IP-10 and IL-15 in serum. Relative to healthy controls, low levels of IL-4, IL-1RA, GRO and VEGF, and high levels of IL-2 in nasal lining fluid, were confined to PCR-confirmed COVID-19 and PCR−/IgG+ SARI participants (Fig. 2). While in serum, low levels of EGF, Flt-3L, MDC and IL-12p70 in serum were distinctive to PCR-confirmed COVID-19 and PCR-/IgG+ SARI participants. In addition, IL-3 levels were distinctively high in nasal lining fluid and serum of PCR-confirmed COVID-19 and PCR−/IgG+ SARI participants, relative to healthy controls (Fig. 2 and Supplementary Figs. 2 and 3). The results demonstrate SARS-CoV-2 infection induces a distinct cytokine response in the nasal mucosa compared to systemic circulation. This also suggests that PCR-confirmed COVID-19 and PCR−/IgG+ SARI participants may be presenting at different stages of the SARS-CoV-2 infection spectrum.

**Severe COVID-19 is associated with a distinct nasal and serum cytokine profile to non-COVID-19 SARIs.** Due to the similarity of the cytokines induced in PCR−/IgG+ SARI participants relative to PCR-confirmed COVID-19 (Fig. 2), coupled with clinical manifestations of COVID-19 (Table 1), and the positive SARS-CoV-2 IgG serology result (Fig. 1b), we explored if clinical reclassification of PCR−/IgG+ SARI participants as COVID-19 disease was warranted. We hypothesised that using an unsupervised analysis the immunological phenotype of PCR−/IgG+ SARI participants would be analogous to PCR-confirmed COVID-19 participants, but distinct from PCR−/IgG− SARI participants and healthy controls. First, we performed a principal component analysis (PCA) of all the analytes for the patients and controls in either nasal lining fluid and serum. In both compartments, PCA segregated together PCR-confirmed COVID-19 participants and PCR−/IgG+ SARI participants, away from healthy controls, whereas PCR−/IgG− SARI participants showed major overlap with the other groups (Fig. 3a, b). The cytokines that contributed to the clustering of the study groups away from the controls were IL-6, IL-4, IL-3, MIP-alpha, G-CSF, IL-1beta in nasal samples and IL-3, IL-6, Flt-3L, EGF, IFN-gamma, IL-5, IL-12p70, IP-10, IL-10, IL12p40 in serum. Second, we analysed the active cytokine functional families in nasal and serum, to ascertain whether PCR−/IgG+ SARI participants had immunological pathways similar to PCR-confirmed COVID-19 participants. Analysis of the active cytokine functional families has previously been used to define immune signatures unique to COVID-19[10–12]. To aid interpretation of the data, the cytokines were divided into functional groups, including growth factors, chemokines, adaptive, pro-inflammatory and anti-inflammatory. In both nasal lining fluid and serum, differential interaction of cytokines was observed among the study groups, with the cytokine interaction profile being generally similar between PCR−/IgG+ SARI participants and PCR-confirmed COVID-19 participants, but distinct from PCR−/IgG− SARI participants and healthy controls (Fig. 3c, d). Specifically, in the nasal lining fluid, we observed a strong interaction in the chemokine family, including between chemokines and growth factors, in PCR-confirmed COVID-19 participants and PCR−/IgG+ SARI participants (Fig. 3c). In the serum, fewer interactions were

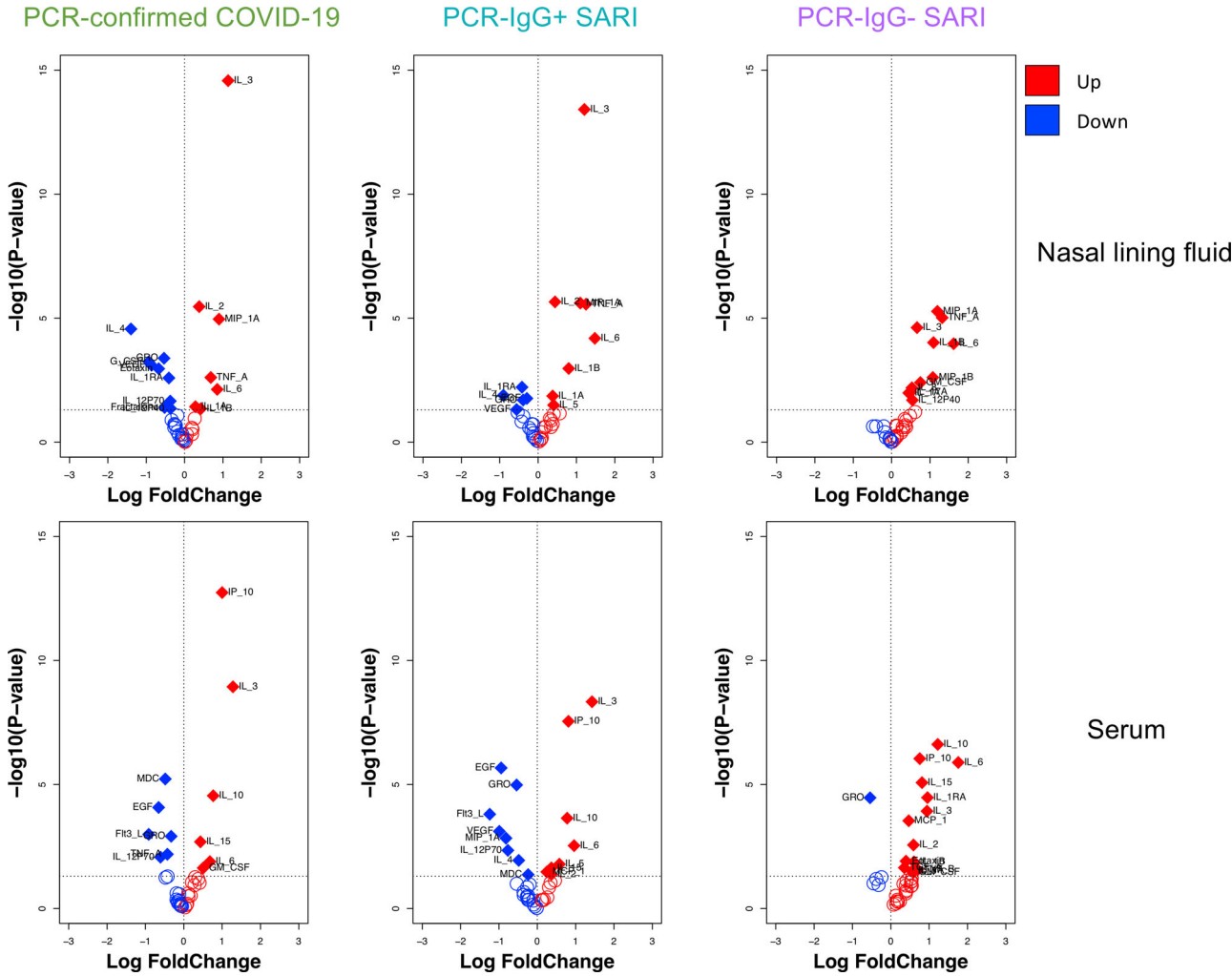

**Fig. 2 Cytokine concentrations in nasal lining fluid and serum.** Volcano plots showing differential cytokine concentrations in nasal lining fluid and serum of PCR-confirmed COVID-19, PCR−/IgG+ SARI and PCR−/IgG− SARI patients compared to health controls. The horizontal dotted line represents a cut-off for statistical significance, while the vertical dotted line represents a cut-off point for determining whether the levels of the cytokines were higher (right, red) or lower (left, blue) compared to healthy controls. Data were analysed using empirical Bayes moderated two-sided *t*-tests and adjusted *p* values are reported (healthy controls, $n = 25$; PCR-confirmed COVID-19, $n = 25$; PCR−/IgG+ SARI, $n = 16$; PCR−/IgG− SARI, $n = 11$). SARS-CoV-2 severe acute respiratory syndrome coronavirus 2, COVID-19 coronavirus disease of 2019, PCR polymerase chain reaction, IgG immunoglobulin G, SARI severe acute respiratory infection. Source data are provided as a Source Data file.

observed, but still a distinct pattern in the adaptive cytokine family in PCR-confirmed COVID-19 participants and PCR−/IgG+ SARI participants (Fig. 3d). Third, in a subset of participants, we explored the nasal immune cellular profile in PCR-confirmed COVID-19 and PCR−/IgG+ SARI participants compared to healthy controls using flow cytometry (gating strategy, Supplementary Fig. 4). We observed a higher frequency of neutrophils and lower frequency of CD3$^+$ T cells in the nasal mucosa of PCR-confirmed COVID-19 and PCR−/IgG+ SARI participants compared to healthy controls (Fig. 4). Collectively, the findings demonstrate that the PCR−/IgG+ SARI participants induced similar immunological pathways to PCR-confirmed COVID-19 participants, but that this was distinct from PCR−/IgG− SARI participants. Taken together, these findings suggest that clinical reclassification of COVID-19 status could be warranted.

**PCR-confirmed COVID-19 participants exhibit reduced propensity for bacterial colonisation.** Due to the differential clinical management pathways experienced by the PCR-confirmed

COVID-19 and PCR−/IgG+ SARI participants in the study, including beta-lactam antibiotic usage and steroid treatment, we sought to determine the presence of other respiratory pathogens in NP/throat swabs. Bacterial co-infection with SARS-CoV-2 is associated with adverse outcomes[21,22]. A total of 80 participants from the patient groups (from whom we had complete clinical and antibody data) were tested for the presence of respiratory pathogens using the fast-track diagnostics (FTD)-33 respiratory panel. The median time from admission to NP/throat swab collection for the respiratory panel testing was 3 days (IQR 1–5). Using the FTD-33 panel we identified 12 other respiratory pathogens present in our patient population, in addition to SARS-CoV-2 and HIV (Fig. 5a). Overall, 74% (59/80) of participants had one or more pathogen(s) (Fig. 5a). The presence of bacterial pathogens was common, identified in 74% (59/80), but viral pathogens (excluding HIV) were only present in 14% (11/80) of participants (Fig. 5a and Supplementary Fig. 5a). The most common respiratory pathogen across all study groups was *Klebsiella pneumoniae* (Supplementary Fig. 5a). A total of 27/44 of the participants with a *K. pneumoniae* had received ceftriaxone or amoxicillin within 24 h of admission, suggesting that this was

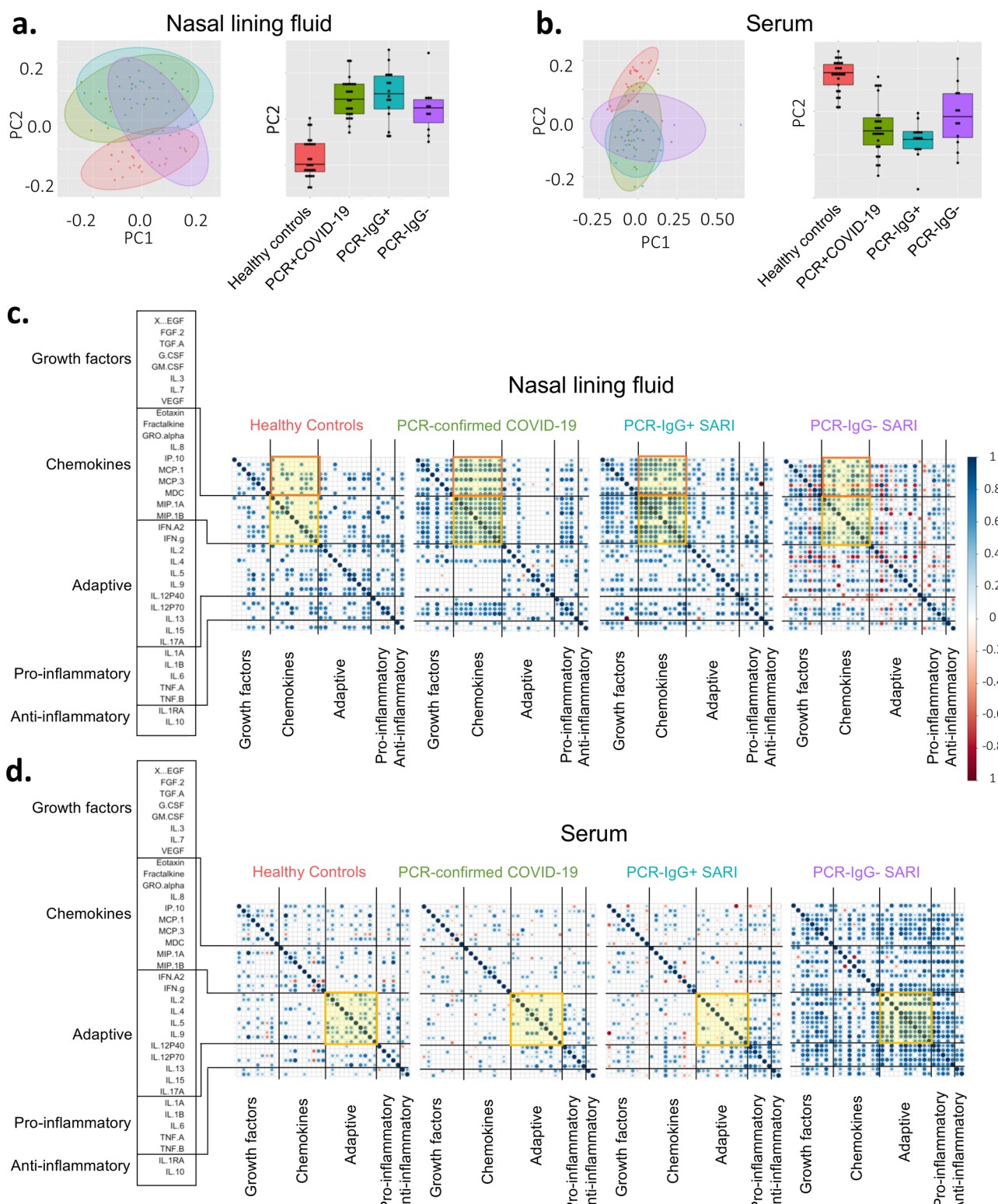

**Fig. 3 Nasal and serum cytokine profiles.** Principal component analysis (PCA) of the 38 analytes (37 cytokines and sCD40L) in **a** nasal lining fluid and **b** serum of healthy controls, PCR-confirmed COVID-19, PCR−/IgG+ SARI and PCR−/IgG− SARI patients, showing similarity of nasal cytokine responses between PCR-confirmed COVID-19 and PCR−/IgG+ SARI patients. Correlogram of cytokine interactions in **c** nasal lining fluid and **d** serum among the different study groups, showing induction of similar immune process in PCR-confirmed COVID-19 and PCR−/IgG+ SARI patients. For all boxplots, box boundaries correspond to 25th and 75th percentiles; whiskers extend to a maximum of 1.5× the interquartile range, with values outside the box and whiskers being outliers. Healthy controls, $n = 25$; PCR-confirmed COVID-19, $n = 25$; PCR−/IgG+ SARI, $n = 16$; PCR−/IgG− SARI, $n = 11$. SARS-CoV-2 severe acute respiratory syndrome coronavirus 2, COVID-19 coronavirus disease of 2019, PCR polymerase chain reaction, IgG immunoglobulin G, SARI severe acute respiratory infection, PC1 principal component 1, PC2 principal component 2. Source data are provided as a Source Data file.

**a.**

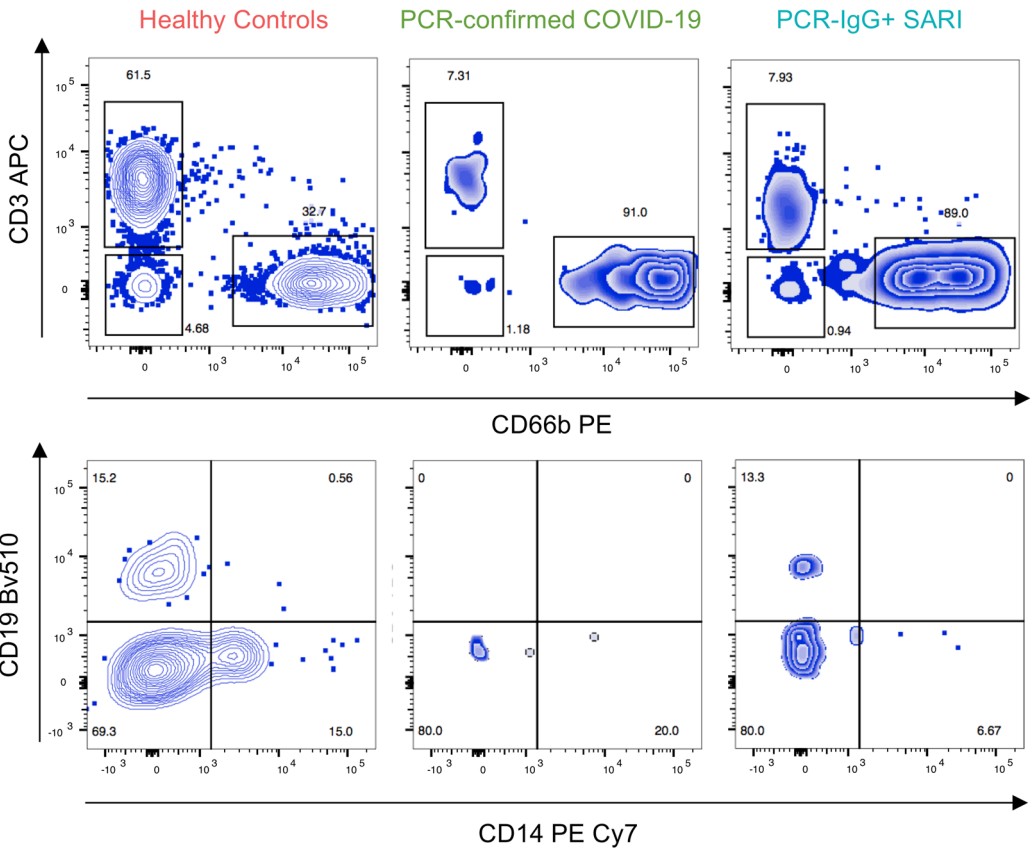

**b.**

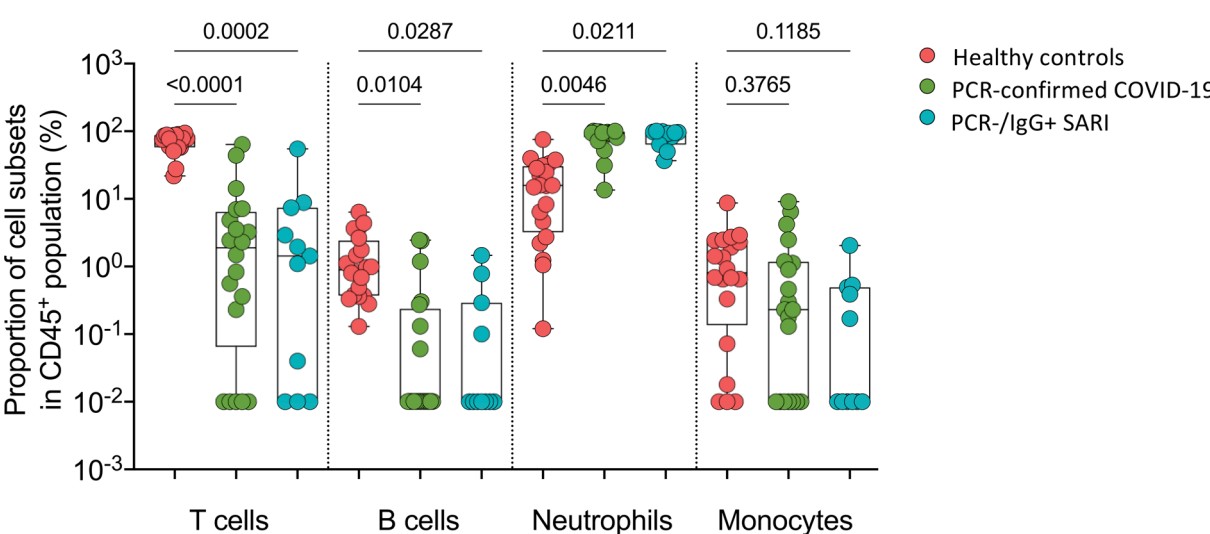

**Fig. 4 Nasal cell composition in healthy controls, confirmed and suspected COVID-19 patients. a** Representative flow cytometry plots for cellular composition in the nasal mucosa of healthy controls, PCR-confirmed COVID-19 and PCR−/IgG+ SARI patients. **b** Proportions of T cells, B cells, neutrophils and monocytes in nasal mucosa of healthy controls and COVID-19 patients. For all boxplots, box boundaries correspond to 25th and 75th percentiles; whiskers extend to a maximum or minimum greatest value. Data were analysed using Kruskal–Wallis test, two-sided (healthy controls, $n = 20$; PCR-confirmed COVID-19, $n = 20$; PCR−/IgG+ SARI, $n = 11$). SARS-CoV-2 severe acute respiratory syndrome coronavirus 2, COVID-19 coronavirus disease of 2019, PCR polymerase chain reaction, IgG immunoglobulin G, SARI severe acute respiratory infection. Source data are provided as a Source Data file.

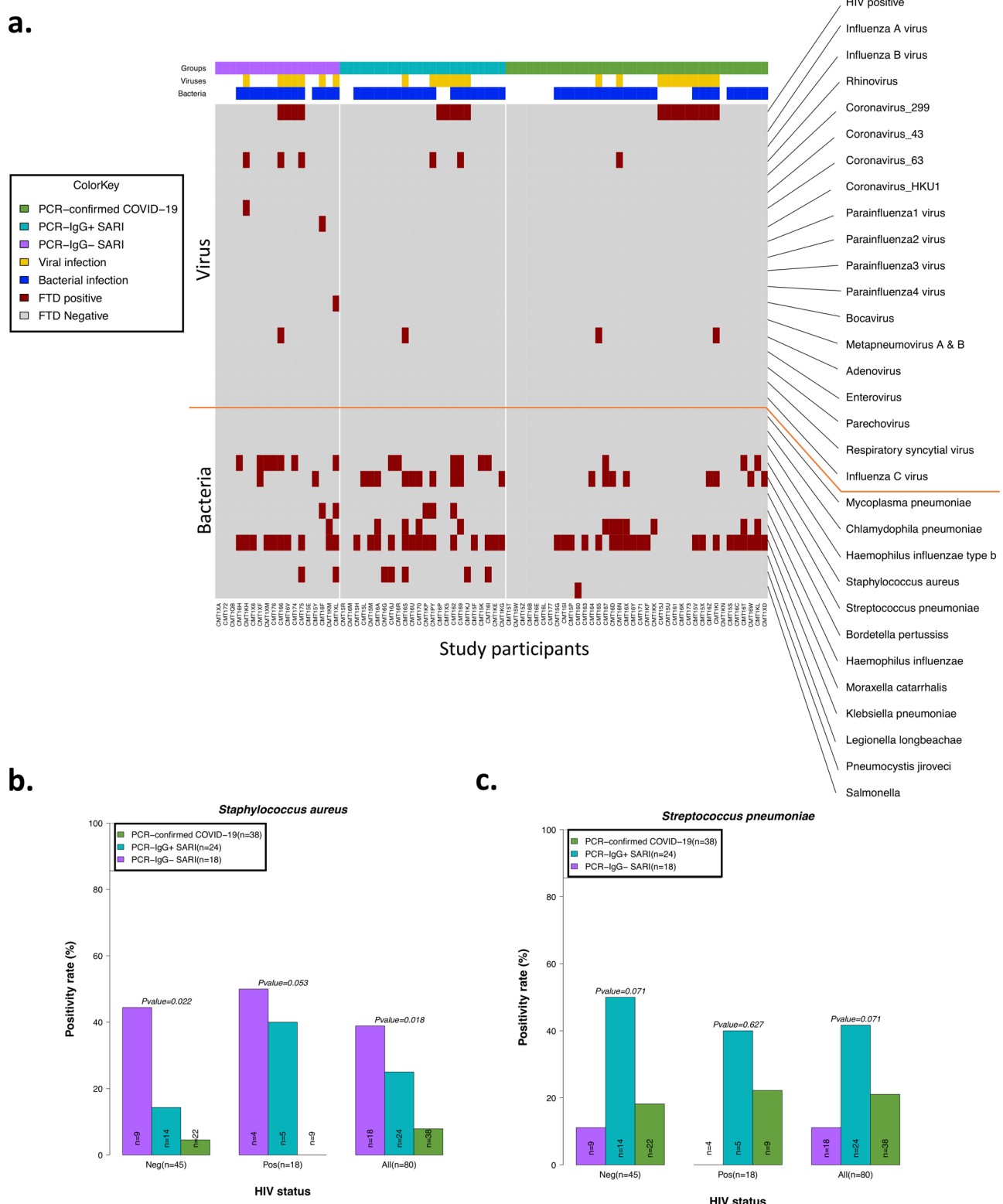

**Fig. 5 Co-colonisation/infection status of suspected and confirmed COVID-19 patients. a** Profile of co-colonisation/infection pathogens in nasopharyngeal/throat swabs. Prevalence of **b** *Staphylococcus aureus* and **c** *Streptococcus pneumoniae* co-colonisation in suspected and confirmed COVID-19. The horizontal bars represent the median and interquartile range (IQR). Data were analysed using Kruskal–Wallis test (PCR-confirmed COVID-19, *n* = 39; PCR−/IgG+ SARI, *n* = 23; PCR−/IgG− SARI, *n* = 22). COVID-19 coronavirus disease of 2019. Source data are provided as a Source Data file.

likely a hospital-acquired multidrug resistant (MDR) *K. pneumoniae*. MDR *K. pneumoniae* is known to be endemic at Queen Elizabeth Central Hospital[23]. Furthermore, we observed differences between PCR-confirmed COVID-19 and PCR−/IgG+

SARI participants. The PCR-confirmed COVID-19 participants had lower prevalence rates of *Staphylococcus aureus* (3/38 [8%] vs. 6/24 [25%], *p* = 0.018) and tended toward lower *Streptococcus pneumoniae* (8/38 [21%] vs. 10/24 [42%], *p* = 0.071)

than PCR−/IgG+ SARI participants (Fig. 5b, c). These findings show that presence of bacterial pathogens in the upper respiratory tract was very common in our hospitalised patient cohort. They also show that PCR-confirmed COVID-19 patients had lower proclivity for *S. aureus* and *S. pneumoniae* colonisation than PCR−/IgG+ SARI participants, potentially impacted by increased use of beta-lactam antibiotic usage in this study group.

**Association of HIV co-infection with cytokine responses, respiratory bacterial prevalence and clinical outcomes in COVID-19 patients.** In our cohort, people living with HIV were distributed across the patient groups (Table 1). We therefore explored the influence of HIV co-infection on the cytokine responses, presence of respiratory pathogens and mortality. We reclassified the PCR−/IgG+ SARI participants as COVID-19 and combined them with the PCR-confirmed COVID-19 to form a single COVID-19 patient group. There were no statistically significant differences in the levels of cytokines in nasal lining fluid and serum between HIV-infected and HIV-uninfected COVID-19 participants (Fig. 6a, b). Furthermore, we did not observe increased frequency of respiratory pathogens in NP/ throat swab nor higher mortality (Fisher's exact test, $p = 0.277$) in

HIV-infected compared to HIV-uninfected COVID-19 participants (Fig. 5b, c and Supplementary Fig. 5b). Together, in our cohort, we did not find evidence suggestive that HIV co-infection in COVID-19 participants was associated with altered cytokine responses, increased prevalence of respiratory pathogens nor increased mortality. It is important to state that our study was not explicitly powered to detect differences between HIV-infected and HIV-uninfected individuals and that this is an exploratory analysis that requires validation in future studies.

**Discussion**

We provide clinical and immunological analysis of suspected or confirmed COVID-19 patients admitted to hospital in Malawi, a low-income sub-Saharan African country. We identified individuals with SARI and manifestations of COVID-19 who were SARS-CoV-2 RT-qPCR negative but IgG positive that showed immunological profile analogous to PCR-confirmed COVID-19 participants. This was distinct from PCR−/IgG− SARI participants and healthy controls. The PCR−/IgG+ SARI subgroup experienced poorer clinical outcomes compared to PCR-confirmed participants, potentially due to later hospital presentation, failure to access effective treatment for COVID-19 or superadded bacterial infections. In our cohort, access to drugs such as dexamethasone in patients suspected of COVID-19 was determined by RT-qPCR test positivity and not SARS-CoV-2 IgG antibody status. Identification of this subgroup of SARI patients highlights the importance for further optimisation of triage and clinical treatment pathways in the era of COVID-19, especially in low-resource settings.

Low cost, effective and pragmatic interventions such as dexamethasone have been widely adopted and incorporated into treatment pathways[24]. The COVID-19 treatment guidelines recommend against dexamethasone administration for patients who do not require supplemental oxygen[25] but there are no specific recommendations for how COVID-19 should be diagnosed. The RECOVERY dexamethasone trial[24] that showed efficacy in hospitalised COVID-19 patients included patients with "clinically suspected" or laboratory confirmed SARS-CoV-2 infection. The PCR−/IgG+ SARI subgroup could potentially fit into this "clinically suspected" COVID-19 classification. Empirical antibiotics are recommended for severe COVID-19 patients if co-existing bacterial pneumonia cannot be excluded[26]. Consistent with this recommendation, there was increased beta-lactam antibiotic usage in PCR-confirmed COVID-19 participants who demonstrated a lower prevalence of *S. pneumoniae* and *S. aureus* colonisation. The PCR−/IgG+ participants did not access these standardised clinical management strategies and may be at a disadvantage compared to PCR-confirmed patients, with whom they share analogous immunological profiles, potentially putting the patients at increased risk of poor prognosis. Well conducted interventional trials are required to determine if this important subgroup could benefit from diagnostic reclassification and treatment as COVID-19.

Cytokine dysregulation is a hallmark of severe COVID-19 (ref. [27]). Consistent with published studies[5,6], we observed high concentrations of inflammation-associated cytokines, including IL-6, TNF-α, IP-10, IL-10, IL-1α and IL-1β, in serum and nasal lining fluid from PCR-confirmed COVID-19 patients. However, elevation of these cytokines was also observed in serum from non-COVID SARI participants. Instead, the induction of the chemokine family in the nasal mucosa was distinctive to severe COVID-19 and was distinct from systemic circulation. The predominance of the chemokine family was in line with the infiltration of neutrophils in the upper airway observed in this cohort and others[28,29]. A study that performed transcriptomic analysis

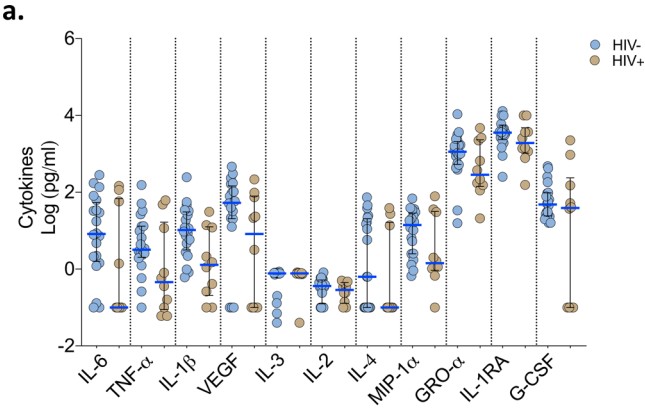

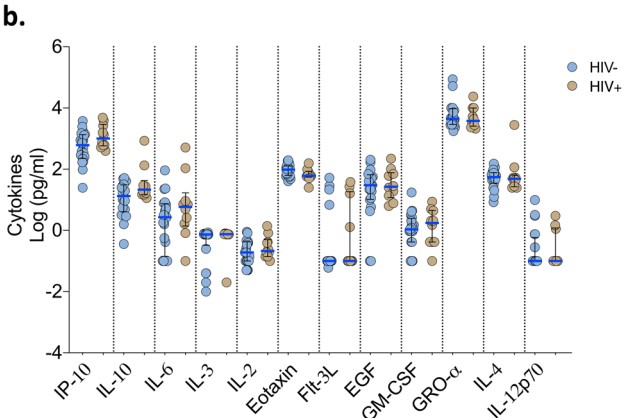

**Fig. 6 Cytokine concentrations in nasal lining fluid and serum based on HIV-co-infection status.** PCR-confirmed COVID-19 and PCR−/IgG+ SARI patients were combined ($n = 41$) into one COVID-19 patient group. Only cytokines that showed differential responses between patients with healthy controls were included in the analysis. **a** Cytokine concentrations between HIV-infected and uninfected COVID-19 patients in nasal lining fluid. **b** Cytokine concentrations between HIV-infected and uninfected COVID-19 patients in serum. The horizontal bars represent the median (blue line) and interquartile range (IQR) (black lines). Data were analysed using Kruskal–Wallis test (HIV−, $n = 20$; HIV+, $n = 10$). COVID-19 coronavirus disease of 2019, PCR polymerase chain reaction, SARI severe acute respiratory infection. Source data are provided as a Source Data file.

on paired upper and lower respiratory tract samples from two COVID-19 patients and showed very high congruency in the majority of the cell types between the sites[28], suggesting a shared immune response between the sites. Consistent with this suggestion, the chemokine-dominated signature and neutrophil infiltration observed in the upper airway of severe COVID-19 patients in our study was similar to that reported in the lower airway[15,16,30]. Neutrophils promote inflammation and play a pathogenic role in COVID-19 (ref. [31]). Circulating neutrophils from severe COVID-19 patients show exaggerated oxidative burst, NETosis and phagocytosis relative to healthy controls[32]. Measurement of dysregulated neutrophil function in the nose was not possible in this study, but presence of NETs has been reported in the lungs of deceased severe COVID-19 patients with acute respiratory distress syndrome (ARDS)[33], where they are thought to drive severe pulmonary complications of COVID-19. In ARDS patients, low levels of VEGF in the lower airway are a marker of acute lung injury[34], and in our study, severe COVID-19 patients exhibited distinctively low concentrations of VEGF in nasal lining fluid. Collectively, this suggests that further investigation to determine if the nasal mucosa could provide a snapshot of immunological activity in the lung in patients with COVID-19 is warranted, as nasal sampling is well tolerated and more easily accessible than lower airway sampling.

Some of the major features of severe COVID-19 are lymphopenia and neutrophilia in the systemic circulation[35,36]. As such, the neutrophil to lymphocyte ratio has been shown to have prognostic value, predicting those at high risk of severe disease or death[37]. However, the mechanisms behind lymphopenia and neutrophilia during COVID-19 are still not well understood. Interestingly, in our study, concentrations of IL-3 and Flt-3L were distinctively altered in severe COVID-19 participants compared to non-COVID-19 SARI participants. Specifically, in severe COVID-19 participants, IL-3 was detected at very high levels in the nasal lining fluid, while Flt-3L was repressed in serum. IL-3 and Flt-3L are key cytokines involved in haematopoiesis of leucocytes[38,39]. IL-3 induces expansion and generation of myeloid cells[38,39], while Flt-3L binds to Flt-3 and activates common lymphoid progenitor cells to increase the number of lymphocytes[40,41]. Alterations in these cytokines in mouse models are associated with dysregulated haematopoiesis and altered leucocyte cellularity[39,41]. Therefore, it is plausible that high levels of IL-3 and repressed levels of Flt-3L could contribute to the high neutrophil to lymphocyte ratio and altered leucocyte numbers observed in severe COVID-19 patients.

Due to the high HIV prevalence in our setting, we conducted an exploratory analysis of the impact of HIV co-infection on COVID-19. While data from the United Kingdom and South Africa suggests an increased risk of mortality in HIV-infected COVID-19 patients[42–44], a systematic review and data from the United States did not demonstrate this association[45–47]. Our cohort was not powered to detect differences in groups by HIV status and we did not find statistically significant differences in the cytokine responses, prevalence of respiratory pathogens nor mortality between HIV-infected and HIV-uninfected COVID-19 participants. The small sample size of this study limits our ability to adjust for the multiple potential confounders that could impact on clinical outcomes, including age, gender, co-morbidities, antiretroviral therapy status and HIV viral load. Larger cohort studies with comprehensive immunological data and sufficient power to adjust for multimorbid diseases are required to provide further clarity on this issue.

Despite the strengths of this study including use of an internationally recognised protocol with standardised data collection tools, well-characterised clinical cohort and paired nasal and systemic immune responses, our study had some limitations. Due to programmatic constraints, it was not feasible to conduct longitudinal sampling among our participants to monitor recovery and response to therapy. In addition, while the majority of participants were recruited within the first 72 h of admission, a proportion of our participants was recruited later in their hospital admission. This is an important limitation because immunological parameters are known to change during the course of disease and may have been confounded by steroid or antibiotic use. Our sample size precluded detailed analysis to examine for confounders. In particular, there were significant differences in age between the three groups. Lack of critical care facilities precluded universal recruitment and sampling in the most severe cases, and our patient population may therefore not be entirely representative of all participants with SARS-CoV-2-induced SARI. Furthermore, we cannot exclude secondary bacterial infections as potential contributors to increased mortality in the PCR−/IgG+ SARI participants, as we did not have access to blood culture or autopsy results.

We have demonstrated that SARS-CoV-2 infection induces a chemokine and neutrophil-dominated profile in the nasal mucosa different from systemic circulation, and distinct from non-COVID-19 SARI. We have identified a subgroup of SARS-CoV-2 PCR-negative IgG-positive individuals with clinical and immunological manifestations of COVID-19, who may benefit from standardised COVID-19 clinical management protocols (including use of steroids and beta-lactam antibiotics). Further, operationalisation of sensitive and specific antibody assays for SARS-CoV-2 IgG detection to support clinical diagnosis needs to be considered in resource-limited settings. We recommend that interventional trials should target this clinically important subgroup of patients to determine if treatment pathways applied for PCR-confirmed COVID-19 can be implemented safely for PCR−/IgG+ SARI patients to improve clinical outcomes.

## Methods

**Study design and recruitment**. We prospectively recruited patients using the tier one sampling strategy from the International Severe Acute Respiratory and Emerging Infection Consortium (ISARIC) Clinical Characterisation Protocol (CCP)[48]. Briefly, the CCP is a standardised protocol that enables data and biological samples to be collected rapidly in a globally harmonised manner for any severe respiratory infection of public health interest[48]. Patients over 18 years old were approached for informed written consent if they met inclusion criteria: SARI with suspected or confirmed SARS-CoV-2. We used WHO case definitions for SARI: history of or measured fever (≥38 °C), cough, onset within last 10 days and requires hospitalisation[49]. For patients who lacked capacity, assent was sought from a proxy as per our ethical approvals. Subsequently, informed consent was obtained retrospectively from these patients, where possible. Patients were excluded from recruitment if they or their proxy declined to participate. We aimed to recruit within 72 h of hospital admission. Nasopharyngeal airway, nasosorption, nasal biopsy and peripheral blood samples were collected at the point of patient recruitment. Thereafter, patients were followed up until hospital discharge or death. All clinical data were collected using REDCap (v9.5.0).

All participants were recruited at Queen Elizabeth Central Hospital (QECH), Blantyre, Malawi. QECH is a government hospital of 1000 beds, providing free inpatient medical care to the city of Blantyre (population 800,264), and tertiary care to those referred from the Southern region (population 7,750,629, 2018). Most adults present directly to the hospital or are referred from community healthcare facilities. After triage and testing in dedicated areas, patients with confirmed SARS-CoV-2 were treated in a cohort ward, including a high dependency area[50]. Additionally, patients with SARI were screened for SARS-CoV-2 in two separate adult medical wards. Context-sensitive standard operating procedures were used to treat SARS-CoV-2 as detailed in a separate publication[51]. Invasive mechanical ventilation, continuous positive airways pressure and high flow oxygen were not available for SARS-CoV-2 at this institution during this period. Clinical data and statistical code are available[52].

During the period November 2019 to October 2020, we recruited healthy participants with no acute intercurrent or chronic illness as a healthy control group. All healthy participants, recruited immediately before (November 2019–March 2020) and after (September 2020–October 2020) the first peak in Malawi's reported COVID case count (Fig. 1a), were confirmed as seronegative for

HIV infection and had no known medical conditions. We incorporated baseline samples for healthy participants who had volunteered to participate in another study in this analysis[53].

We have complied with all relevant ethical regulations for work with human participants, and written informed consent was obtained for all participants. The two study protocols were approved by the Malawi National Health Science Research Committee (NHSRC, 20/02/2518 and 19/08/2246) and Liverpool School of Tropical Medicine (study sponsor) Research Ethics Committee (LSTM REC, 20/026 and 19/017). Patient and health participant samples were anonymised at the point of sample collection by the research nurses using unique participant identification barcodes. Study activities were monitored by the Malawi-Liverpool-Wellcome Trust Clinical Research programme (MLW)'s Clinical Research Support Unit and we complied with all relevant ethical regulations. Approval for the SARS-CoV-2 ELISA verification samples was obtained from the College of Medicine Research Ethics Committee (COMREC, P.05/20/3045).

**SARS-CoV-2 PCR diagnostic testing.** All participants underwent SARS-CoV-2 testing at hospital admission, and subsequently after study recruitment. After collection, nasopharyngeal swabs in Universal Transport Medium (UTM) (Copan, Brescia, Italy) were stored at 7 °C and processed for 2019-nCOV RNA testing within 48 h, using the CDC 2019-nCoV RNA real-time reverse transcriptase PCR diagnostic panel (Integrated DNA Technologies, Iowa, USA) or the Da An-RT-PCR reagent set for 2019-nCoV RNA detection (Da An Gene Co., Ltd of Sun Yat-Sen University, Guangdong, P.R. China). A cycle threshold (Ct) value of <40 was considered positive for both assays based on CDC and Da An guidelines using QuantStudio Real-Time PCR software v1.3 (Applied Biosystems, UK). Both assays utilise an internal control to identify presence of human RNA (CDC—ribonuclease Protein, Da An internal control is not published). A negative extraction control and a PCR no-template control were also performed with every test.

**Respiratory fast track diagnostic panel.** Aliquots of UTM were stored at −80 °C and tested in batches using the FTD® Respiratory Pathogens 33 kit (Fast track Diagnostics Ltd, Luxembourg) as per the manufacturer's instructions. In brief, samples were extracted using the QIAamp UCP Pathogen Mini Kit (Qiagen Ltd, UK) for both DNA and RNA. Each sample was then tested using the RT-qPCR based FTD panel. This panel includes the following pathogens: parainfluenza viruses 1, 2, 3 and 4; human coronaviruses NL63, 229E, OC43 and HKU1; human metapneumovirus A/B; human rhinovirus; human respiratory syncytial viruses A/B; human adenovirus; enterovirus; human parechovirus; human bocavirus; *Pneumocystis jirovecii; Mycoplasma pneumoniae; Chlamydophila pneumoniae; Streptococcus pneumoniae; Haemophilus influenzae* B; *Staphylococcus aureus; Moraxella catarrhalis; Bordetella* spp.; *Klebsiella pneumoniae; Legionella pneumophila/longbeachae; Salmonella* spp.; *Haemophilus influenzae* and internal control. Ct values <40 were considered positive. Diagnostic data and code are available[54].

**SARS-CoV-2 IgG enzyme linked-immunosorbent assay.** Peripheral blood collected in serum separation tubes underwent centrifugation at 500g for 8 min to isolate serum. Serum was stored at −80 °C. To measure SARS-CoV-2 antibodies, we used a CE-marked commercial enzyme linked-immunosorbent assay (ELISA) targeting Spike (S2) and Nucleoprotein (NP) from SARS-CoV-2 (Omega diagnostics, UK; ODL 150/10; Lot #103183). This assay has previously been used in our context in a published healthcare worker seroprevalence study[55]. We have also tested this assay on WHO-supplied NIBSC COVID-19 Convalescent Plasma Panel (20/120, 20/122, 20/124, 20/126, 20/128 and 20/130) and found 100% concordant results. The assay was performed as per the manufacturer's instructions. In brief, participant serum was diluted (1:200) in sample diluent (150mM Tris-buffered saline, pH 7.2 with antimicrobial agent). The diluted samples, diluent alone (negative control), manufacturer's cut-off control and positive control were added at 100 μl per well to a plate pre-coated with S2 and NP. The plate was incubated at room temperature for 30 min. After incubation, the plate was washed three times with a wash buffer (100 mM Tris-buffered saline with detergent, pH 7.2) using a plate washer (Asys Atlantis, Biochrom Ltd, UK). One hundred microliters anti-human IgG conjugated to horseradish peroxidase was then added to each well and incubated for 30 min at room temperature. After incubation, the plate was washed four times with a wash buffer, and 100 μl of TMB (3,3′,5,5′-tetramethylbenzidine) substrate (aqueous solution of TMB and hydrogen peroxide) was added. The plate was incubated for 10 min at room temperature, before addition of 100 μl of Stop Solution (0.25 M sulfuric acid). The optical density (OD) of each well was read at 450 nm in a microplate reader (BioTek ELx808, UK) within 10 min of adding the Stop Solution using Gen 5 software v2.09 (BioTek, UK). The ratio of OD in the test samples to the assay threshold control was calculated. The assay interpretation was as follows: positive result (ratio ≥1) and negative result (ratio <1). All serology data are available[56].

**Flow cytometry analysis.** For immunophenotyping, nasal cells were dislodged from curettes by pipetting and stained with an antibody cocktail containing anti-human CD3 APC (2D1, 368514, 1:40) and anti-human CD66b PE (UCHT1, 300439, 1:50) from Biolegend (UK), and anti-human CD45 Alexa Fluor 700 (G10F5, 12-0666-42, 1:80) from eBioscience (UK). Samples were acquired on an LSR FORTESSA flow cytometer using FACSDIVA (BD Biosciences, UK) and analysed using Flowjo v10.5.3 (BD Biosciences, USA). All flow cytometry data are available[56].

**Luminex analysis of nasal lining fluid.** Cytokines were eluted from stored nasosorption filters (Mucosal Diagnostics, Hunt Developments (UK) Ltd, Midhurst, UK) using 200 μl of elution buffer (Millipore) by centrifugation at 1500g, and then the eluate was cleared by further centrifugation at 1595g. The samples were acquired on a MAGPIX (Luminex, UK) using a 38-plex magnetic human cytokine kit (Millipore) and analysed with xPONENT software following the manufacturer's instructions. The analytes included sCD40L, EGF, Eotaxin/CCL11, FGF-2, Flt-3 ligand, Fractalkine, G-CSF, GM-CSF, GRO, IFN-α2, IFN-γ, IL-1α, IL-1β, IL-1RA, IL-2, IL-3, IL-4, IL-5, IL-6, IL-7, IL-8, IL-9, IL-10, IL-12 (p40), IL-12 (p70), IL-13, IL-15, IL-17A, IP-10, MCP-1, MCP-3, MDC (CCL22), MIP-1α, MIP-1β, TGF-α, TNF-α, TNF-β and VEGF. All cytokine data and code are available[56].

**Statistical analysis.** Clinical data were analysed using Stata V15.1 (StataCorp, Stata Statistical Software: Release 15, College Station, Texas, USA). Categorical variables were compared using the Fisher's exact test. Continuous variables were tested for normality and appropriate statistical tests applied. Non-normally distributed measurements are expressed as the median and were analysed by the Kruskal–Wallis test. For the volcano plots, data were analysed using empirical Bayes moderated *t*-tests with adjusted *p* values reported. A *p* value of <0.05 was considered statistically significant. Immunological and diagnostic data and all figures were produced using R v.3.5.1 (R Development Core Team, Vienna, Austria), RStudio (v1.1447), ggplot2 (v3.3.2), ggfortify (v0.4.11), corrplot (v0.84) and GraphPad Prism v9.0.0 (GraphPad Software, San Diego, California, USA).

**Reporting summary.** Further information on research design is available in the Nature Research Reporting Summary linked to this article.

## Data availability

Source data are provided with this paper. Data and code have also been uploaded to a data repository and are freely available at https://dataverse.harvard.edu/dataverse/BlantyreCOVID. We have specifically cited links to clinical[52], diagnostic[54] and immunological[56] data sets and code within the "Methods" section. Source data are provided with this paper.

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

## Acknowledgements

The authors thank all study participants and the staff of the Queen Elizabeth Central Hospital (QECH) for their support and co-operation during the study. This work was supported by the UK Foreign, Commonwealth and Development Office and Wellcome [220757/Z/20/Z] (to B.M.). Further support was provided by Wellcome [211433/Z/18/Z] (to S.B.G.) and National Institute for Health Research (NIHR) [16/136/46] (to K.C.J.). K.C.J. is supported by an MRC African Research Leader award [MR/T008822/1]. J.R. is supported by a Wellcome Trust Career Development Fellowship [211098/Z/18/Z]. K.G.B. is supported by a NIH K-award (TW010853). K.J. is supported by a Wellcome International Training Fellowship [201945/Z/16/Z]. A Wellcome Strategic award number 206545/Z/17/Z supports the Malawi-Liverpool-Wellcome Trust Clinical Research Programme (to S.B.G.). The views expressed are those of the authors and not necessarily those of the UK NHS, the NIHR or the Department of Health and Social Care.

## Author contributions

The author contributions were as follows: Methodology: B.M., K.C.J., K.G.B., J.C., C.V.D.V., J.R. and K.J. Investigation: B.M., K.G.B., C.A., K.J., P.M., J.M., R.K., C.B., J.N., C.V.D.V., N.P.B., T.P., K.S.M., K.M., J.R., C.P., J.M., M.N., G.K., P.K., J.J., H.C.W., S.B.G. and K.C.J. Data analysis: K.C.J., K.G.B., J.J., B.M. and J.C. Interpretation: K.C.J., B.M., K.G.B., J.C., J.J. and S.B.G. Data curation: P.K. Project administration: B.M., K.G.B., C.A., J.C., K.C.J., R.K. Sample collection: B.M. and members of the Blantyre COVID-19 consortium (clinical). Writing: B.M., K.C.J., K.G.B., J.J., J.C. and S.B.G. Conceptualisation and supervision: B.M., K.C.J., K.G.B. and J.C. All authors read and approved the final manuscript.

## Competing interests

The authors declare no competing interests

## Additional information

## Blantyre COVID-19 Consortium

**Clinical** Jacob Phulusa[1], Mercy Mkandawire[1], Sylvester Kaimba[1], Herbert Thole[1], Sharon Nthala[1], Edna Nsomba[1], Lucy Keyala[1], Peter Mandala[1], Beatrice Chinoko[1], Markus Gmeiner[1,2], Vella Kaudzu[1], Samantha Lissauer[1,4], Bridget Freyne[1,4], Peter MacPherson[1,2], Todd D. Swarthout[1,2,10] & Pui-Ying Iroh Tam[1,2]

**Laboratory** Simon Sichone[1], Ajisa Ahmadu[1], Oscar Kanjewa[1], Vita Nyasulu[1], End Chinyama[1], Allan Zuza[1], Brigitte Denis[1], Evance Storey[11], Nedson Bondera[11], Danford Matchado[11], Adams Chande[11], Arthur Chingota[11], Chimenya Ntwea[11], Langford Mkandawire[11], Chimwemwe Mhango[1], Agness Lakudzala[1], Mphatso Chaponda[1], Percy Mwenechanya[1], Leonard Mvaya[1,2] & Dumizulu Tembo[1]

**Data and statistics** Marc Y. R. Henrion[1,2], James Chirombo[1], Clemens Masesa[1] & Joel Gondwe[1]

[10]NIHR Global Health Research Unit on Mucosal Pathogens, Division of Infection and Immunity, University College London, London, UK. [11]Laboratory, Queen Elizabeth Central Hospital, Blantyre, Malawi.

