## [Peer Review File · Nature Communications]

Reviewers' comments:

Reviewer #1 (Remarks to the Author):

This manuscript by Morton and colleagues analyzes clinical and basic immunologic profiles of blood and nasal samples from a cohort of 87 individuals with undefined "severe acute respiratory infection." A portion of this cohort had confirmed COVID-19 either via PCR or Spike/NP ELISA. Furthermore, the authors provide extensive analysis of a contemporaneously-collected cohort of individuals without laboratory-confirmed COVID-19 who appear to have very different immunologic profiles – most likely because most did not have COVID-19. They collected serum and nasopharyngeal swabs as well as nasosorption filters from these individuals at a single time-point several days into the hospital course (median reported for the nasopharyngeal swabs in that section of the results is 3 days) and performed comprehensive 38-plex cytokine analysis and very basic flow cytometry results on a select number of the individuals with nasopharyngeal swab collection. While the peripheral blood data provided are largely repetitive of many data sets that have been previously published, the authors do emphasize their nasopharyngeal sample analysis that is less well-described in the COVID-19 literature. Furthermore, the high HIV seroprevalence in the cohort and the unique geographic location of the cohort make the information that is provided of more interest to the general reader.

I suggest the following points of revision:

1) The clinical definition of "severe acute respiratory infection" is not delineated in the results section or the methods. Please provide. Unfortunately, the supplementary figures were not provided for review and I note that the first supplementary figure is about the collection and cohort – therefore, if the definition of "severe acute respiratory infection" is located there, it was not available to this reviewer. Even so, it should be moved to the first section of the methods. Did these patients meet any additional criteria other than the fact that they had symptoms of a respiratory infection and required admission to hospital? Were they hypoxic? To what degree?

2) Please report the median and IQR for the time from hospital admission to sample collection in Table 1.

3) The analyses performed, the writing and many individual points that are made in the second paragraph of the results section entitled "Distinct cytokine responses induced by SARS-CoV-2 infection in nasal..." are quite confusing and need to be clarified. In the second sentence "(nasal, n=0; serum, n=2)" – does this actually refer to an experiment? You had no samples from a group that you were comparing to other groups?

Further, the entire purpose of this paragraph and the analysis presented in Figure 2b appears to be to show that the individuals with PCR-confirmed vs. IgG+-only-confirmed SARS-CoV-2 both have COVID-19. The last paragraph states this conclusion directly and the point is self-evident. I don't find this analysis useful to the reader.

Finally, Figure 2B is not clear what is being presented. The number of cytokines that are higher or lower in each of the groups in the Venn diagram do not make a lot of sense. Are these the number that are significantly higher or lower than healthy controls? Or is this just the number with a magnitude that is higher or lower than healthy controls? The figure legend alludes to a statistical method here, but it is not clearly described.

Overall, this reviewer found that Figure 2b and the discussion surrounding this figure added very little to the manuscript. It either requires significant revision to emphasize why this analysis is important and improves upon what is shown with the volcano plots in Figure 2a, or it should be removed.

4) The font size in portions of Figures 3c, 3d, 5a and 5b is too small and illegible due to this size.

5) The HIV analysis in Figures 5b, 5c and 6 is critical and justifies further elaboration in the manuscript.

6) 3rd paragraph of the discussion, second to last sentence... "Collectively, this suggests that the nasal mucosa could provide a snapshot of immunological activity in the lung in patients with COVID-19." This is overstated and not supported by the paucity of the observations discussed nor by any data presented in this manuscript. Exclude this statement.

Reviewer #2 (Remarks to the Author):

The authors here describe an interesting study of why we should consider monitoring patients who are PCR-/IgG+, as they could be more susceptible to severe COVID-19. While the sample numbers are generally small, these insights could form the basis for future studies to look into this subset of patients. However, there are several concerns with regards to the interpretation and presentation of data. Specific questions are as follows:

1. Can the authors clarify what the PCR-IgG- SARI group consist of? Are these patients exposed to other viruses?
2. Figure 2a. Increasing lines of evidence has shown that gene and cytokine expression levels can vary with time-point relative to respiratory nadir. The authors should thus describe the time-points in which the samples were taken. Also, would demographics such as age be a plausible confounding factor in the interpretation of data? The demographics of the healthy control should thus be presented as well. Finally, are the p-values adjusted? If so, please state the test performed.
3. Figure 2b. The heatmap is not informative, as this did not reflect the intensity of the cytokine changes in the different groups. Moreover, it is unclear from the map which of the cytokines showed statistical difference. Finally, the venn diagram should indicate the criterion used for comparisons. Was there a fold-change or p-value cut-off?
4. Figure 3. The authors claim that PCA analysis reveal that the PCR-IgG- appeared as a heterogeneous group, but the data did not support the claim. From the graphs, only the healthy controls appeared different from the other groups. Perhaps the authors should consider Partial Least Square methods to better segregate the data. At present, I am unconvinced that the PCA separated the groups apart.
5. Figure 4b. The reviewer note that these subsets were measured based on CD45+ cells, which are the activated cells. Are the overall numbers of these cell subsets changed as well?
6. In the discussion, the authors should highlight some limitations of the study.

Reviewer #3 (Remarks to the Author):

Review for Nature Communications
March 11, 2021

Title: In depth analysis of patients with severe SARS-CoV-2 in sub-Saharan Africa demonstrates distinct clinical and immunological profiles

The study team evaluated both suspected and confirmed COVID-19 patients in Malawi, and compared then to a small group of healthy controls. Overall, they found that those people who were suspected for COVID, but had PCR negative tests were more likely to have other bacterial infections and a higher mortality rate.

The major limitation of this study is the lack of PCR confirmation for many people who were suspected of COVID-19, either by bronchoscopy and/or serial PCR or antigen testing. The finding that COVID-19 participants were more likely to survive compare to PCR-negative patients with severe acute respiratory infection is not a novel finding. The estimated case fatality rate has been well documented, and bacterial infections (which were present for other patients) have a higher case fatality rate. Therefore, without further characterizing the underlying infection in the control group, these comparisons provide little relevance.

The use of confirmatory antibody testing for acute COVID is not typical, and should not be recommended more broadly. While it is possible that IgG may be detectable as early as the first 5-7 days after infection, testing IgG levels should not be part of the diagnostic algorithm for acute infection. In this study, those patients who were considered as COVID+ with a negative PCR test were likely to either be (1) previously infected, but had cleared the virus, or (2) not infected with SARS-cov-2 and had a false antibody test.

While the study did not find evidence that HIV and COVID co-infection was associated with an increased risk of mortality, the study was not powered to detect this association. There were only 9 people living with HIV in each of the 3 groups, so there was little chance for achieving statistical power. Therefore, this statement is misleading and should be removed from the abstract.

While this study presents a lot of lab generated data, there is little data and few associations that are meaningful and significant (despite an excess of figures). Analyzing a large number of cytokines from both a nasal swab and from blood will certainly generate some differences, but those differences may or may not be clinically relevant, and would have been expected if people were in different stages of infection.

This manuscript would have been more informative to focus on the comparisons b/n COVID PCR+ patients against the healthy control group.

Reviewer one

1. *The clinical definition of “severe acute respiratory infection” is not delineated in the results section or the methods. Please provide. Unfortunately, the supplementary figures were not provided for review and I note that the first supplementary figure is about the collection and cohort – therefore, if the definition of “severe acute respiratory infection” is located there, it was not available to this reviewer. Even so, it should be moved to the first section of the methods. Did these patients meet any additional criteria other than the fact that they had symptoms of a respiratory infection and required admission to hospital? Were they hypoxic? To what degree?*

Response: We have now included our definitions of severe acute respiratory infection (WHO case definition, lines 329-330). We have described the proportion of patients with hypoxia who required oxygen in Table 1. The supplementary figures were submitted with the initial submission.

2. *Please report the median and IQR for the time from hospital admission to sample collection in Table 1.*

Response: We have now included information on time from hospital admission to recruitment in Table 1.

We also note an error in the p values for Fisher’s exact tests which had not been updated from incorrectly applied Chi² tests during initial analyses. We have now updated the p values. These updates do not change any of the reported significant variables, nor change the message of the manuscript.

3. *The analyses performed, the writing and many individual points that are made in the second paragraph of the results section entitled “Distinct cytokine responses induced by SARS-CoV-2 infection in nasal...” are quite confusing and need to be clarified. In the second sentence “(nasal, n=0; serum, n=2)” – does this actually refer to an experiment? You had no samples from a group that you were comparing to other groups?.....Further, the entire purpose of this paragraph and the analysis presented in Figure 2b appears to be to show that the individuals with PCR-confirmed vs. IgG+-only-confirmed SARS-CoV-2 both have COVID-19. The last paragraph states this conclusion directly and the point is self-evident. I don’t find this analysis useful to the reader.*

Response: The numbers reflect the number of cytokines whose levels were significantly different from healthy controls, they are not number of samples. In the venn diagrams the numbers represent the number of cytokines that are significantly higher or lower than healthy controls. On reflection, to improve clarity, we have removed this part of the paragraph and also removed Figure 2b. This does not change the message in the manuscript as eluded to by the reviewer.

4. *The font size in portions of Figures 3c, 3d, 5a and 5b is too small and illegible due to this size.*

Response: Font sizes have been adjusted accordingly

5. *The HIV analysis in Figures 5b, 5c and 6 is critical and justifies further elaboration in the manuscript.*

Response: We have provided further elaboration on the HIV analysis within the manuscript whilst being mindful of the limitation in study power to detect associations (lines 210-212).

6. *3rd paragraph of the discussion, second to last sentence... "Collectively, this suggests that the nasal mucosa could provide a snapshot of immunological activity in the lung in patients with COVID-19." This is overstated and not supported by the paucity of the observations discussed nor by any data presented in this manuscript. Exclude this statement.*

Response: We have revised this statement to capture the reviewers concerns (lines 264-267).

Reviewer two

1. *Can the authors clarify what the PCR-IgG- SARI group consist of? Are these patients exposed to other viruses?*

Responses: Figure 5a describes in detail the other pathogens present in the nasopharyngeal samples from all of the study participants, including PCR-/IgG- individuals. The PCR-IgG- SARI group consisted of patients presenting with symptoms consistent with a SARI, but did not have evidence of SARS-CoV-2 at diagnosis. We are not powered to ascertain the aetiological agent for these SARIs in our cohort, suffice to note that there was a trend to increased rhinovirus infection in the PCR-/IgG- group and that we identified two betacoronaviruses, one bocavirus and one adenovirus in this group.

2. *Figure 2a. Increasing lines of evidence has shown that gene and cytokine expression levels can vary with time-point relative to respiratory nadir. The authors should thus describe the time-points in which the samples were taken. Also, would demographics such as age be a plausible confounding factor in the interpretation of data? The demographics of the healthy control should thus be presented as well. Finally, are the p-values adjusted? If so, please state the test performed.*

Response: We had presented the demographics of the healthy control group in Table 1. We have now described the time points at which samples were taken in Table 1 in line with reviewer one comments. We agree about the risk of confounding with particular factors such of age (again described in Table 1), however, we are mindful of our limited sample size and have now added this as a limitation (lines 300-303). Analysis for the data reported in Figure 2a was done

using empirical Bayes moderated t-tests as was indicated in the figure legend and the volcano plot reflects adjusted p values. We have now added this information the statistics section (lines 431-432).

3. *Figure 2b. The heatmap is not informative, as this did not reflect the intensity of the cytokine changes in the different groups. Moreover, it is unclear from the map which of the cytokines showed statistical difference. Finally, the venn diagram should indicate the criterion used for comparisons. Was there a fold-change or p-value cut-off?*

Response: We have removed Figure 2b in line with reviewer one and reviewer two comments.

4. *Figure 3. The authors claim that PCA analysis reveal that the PCR-IgG- appeared as a heterogeneous group, but the data did not support the claim. From the graphs, only the healthy controls appeared different from the other groups. Perhaps the authors should consider Partial Least Square methods to better segregate the data. At present, I am unconvinced that the PCA separated the groups apart.*

Response: We have removed the word heterogeneous to reduce the chance of confusion (Line 149-150). What we were trying to say is that the PCR-IgG- group did significantly cluster separately away from all the study groups including controls, especially in nasal lining fluid (Figure 3a). The potential explanation was that this group was a heterogeneous group due to different aetiological agents. We base our statement that the PCR-/IgG- negative patients are a heterogeneous group based on collective evidence from Figures 2 and 3. Specifically, the volcano plots, PCA and correlograms.

5. *Figure 4b. The reviewer note that these subsets were measured based on CD45+ cells, which are the activated cells. Are the overall numbers of these cell subsets changed as well?*

Response: CD45 (leucocyte common antigen) is a pan leucocyte marker. We have used CD45 to identify leucocytes, separating them from other non-white blood cells including epithelial cells and stromal cells.

6. *In the discussion, the authors should highlight some limitations of the study.*

Response: We agree that there are limitations to this study and had described these within a specific paragraph within the discussion. We have now extended this section, including limitations highlighted by all of the reviewers (lines 295-308).

Reviewer three

1. *The major limitation of this study is the lack of PCR confirmation for many people who were suspected of COVID-19, either by bronchoscopy and/or serial PCR or antigen testing. The finding*

that COVID-19 participants were more likely to survive compare to PCR-negative patients with severe acute respiratory infection is not a novel finding. The estimated case fatality rate has been well documented, and bacterial infections (which were present for other patients) have a higher case fatality rate. Therefore, without further characterizing the underlying infection in the control group, these comparisons provide little relevance.

Response: All patients in our prospective cohort study had their SARS-CoV-2 status confirmed by PCR. This test was conducted sequentially both at hospital admission and subsequently upon sampling after study consent. I'm afraid that reviewer three is incorrect in their assertion on this point as is clearly described in the manuscript. We have now provided increased clarity on this point in the methods section (lines 363-364).

Furthermore, we are anxious that the reviewer has not considered the context in which the patients were treated and samples acquired. Malawi is a low income country where mechanical ventilation for severe COVID-19 is not available. This situation is common to many sub Saharan African countries where there is an extreme lack of data guiding clinicians on how to manage their patients. Requesting that there should be PCR confirmation using bronchoscopic samples is therefore not plausible in this context. We had extensively described hospital facilities in lines 337-346 and highlighted the limitation of lack of critical care facilities in our limitations section (lines 303-305) with the original submission.

We strongly disagree with the reviewer on the relevance of our observation that PCR-confirmed patients were more likely to survive than the PCR-negative patients.

Context: As we have stated in the manuscript (line 221-223), in our setting unlike in developed countries access to life-saving standardized COVID-19 care is dependent on a SARS-CoV-2 PCR-positive result. Further, patients mostly present to hospital late, increasing the probability of a false negative nucleic acid amplification test (NAAT) SARS-CoV-2 result. It is therefore highly plausible to have increased mortality in true COVID-19 patients who have a NAAT-negative test, due to lack of standardised care including dexamethasone treatment.

Relevant literature: The RECOVERY dexamethasone trial recruited not only hospitalized patients with laboratory-confirmed SARS-CoV-2 infection but also clinically suspected patients (N Engl J Med 2021; 384:693-704). The clinically suspected patients equally benefited from the dexamethasone as the individuals with laboratory-confirmed SARS-CoV-2 infection.

- 2. The use of confirmatory antibody testing for acute COVID is not typical, and should not be recommended more broadly. While it is possible that IgG may be detectable as early as the first 5-7 days after infection, testing IgG levels should not be part of the diagnostic algorithm for acute infection. In this study, those patients who were considered as COVID+ with a negative PCR*

test were likely to either be (1) previously infected, but had cleared the virus, or (2) not infected with SARS-cov-2 and had a false antibody test.

Response: We agree with the reviewer that antibody testing for COVID is not typical and indeed this is the message/novelty of the manuscript. However, we strongly disagree with the sweeping statement that this *should not be recommended more broadly*, as it totally disregards context-dependent factors. Patients in low income settings frequently present late to hospital during which time their virus may have cleared. However, as we know from SARS-CoV-2 disease course, the immunological sequelae and severe hypoxia frequently occur after viral clearance. Therefore, testing antibody status could be a useful way to determine recent SARS-CoV-2 exposure and together with patient presentation inform clinical decisions. Indeed, reviewer 1 has specifically picked up on this point stating that it is self evident that both PCR+ and PCR-/IgG+ patients have been exposed to SARS-CoV-2 infection. This is vitally important data, as we have highlighted in the manuscript, to potentially improve access to evidence based therapies such as steroids. The PCR-/IgG+ patients did not receive steroids and had higher mortality than the PCR+ patients. Thus measurement of IgG for patients suspected of COVID-19 but who are PCR negative represent a significant vulnerable (increased mortality) sub group who potentially stand to benefit from COVID-19 treatment protocols. This is explained and justified in lines 227-242.

Addressing the point about potential for false positive results, we used the CE-marked and independently tested Mologic/Omega diagnostics SARS-CoV-2 ELISA platform within this study. This assay has undergone rigorous independent validation at the Liverpool School of Tropical Medicine (UK) and St George's University of London (UK). We used this assay in our previously published work testing seroprevalence in healthcare care workers in Malawi (<https://wellcomeopenresearch.org/articles/5-199/v2>). In the current study, we tested pre-pandemic historical samples (2016-2019) for SARS-CoV-2 S2 and NP IgG antibodies to further verify specificity of the ELISA assay, the samples included sera from individuals with other coronaviruses, malaria convalescent sera, sera from people living with HIV and sera from asymptomatic HIV-uninfected adults. We found that in the historical controls 8/153 (94% [91-97]) were SARS-CoV-2 serology positive. These results are shown in in Figure 1b. In addition, we have tested this assay on WHO-recommended NIBSC COVID-19 Convalescent Plasma Panel (20/120, 20/122, 20/124, 20/126, 20/128 and 20/130) and found 100% concordant results. Therefore, the classification of PCR-IgG+ patients as false positives is not supported by the weight of evidence. We have now included additional information to clarify this point in the methods section (lines 388-393) and Figure Legend (Figure 1).

3. *While the study did not find evidence that HIV and COVID co-infection was associated with an increased risk of mortality, the study was not powered to detect this association. There were only 9 people living with HIV in each of the 3 groups, so there was little chance for achieving statistical power. Therefore, this statement is misleading and should be removed from the abstract.*

Response: Reviewer three highlights that we do not have power to detect the association between HIV infection and COVID-19 co-infection. We completely agree with this and indeed had already explicitly stated this in the manuscript (line 289-293). Indeed, we have not performed tests of association within our analysis for this reason. This reviewer statement is in contrast to reviewer one who has specifically asked for increased elaboration of our HIV analysis within the manuscript. We have now included an additional statement in the results section (lines 210-212) highlighting lack of power and that we conducted an exploratory analysis.

4. *While this study presents a lot of lab generated data, there is little data and few associations that are meaningful and significant (despite and excess of figures). Analyzing a large number of cytokines from both a nasal swab and from blood will certainly generate some differences, but those differences may or may not be clinically relevant, and would have been expected if people were in different stages of infection.*

Response: We strongly disagree with the reviewer's assertion on the meaningfulness and significance of our lab generated data. First, this assertion is in contrast to the other two reviewers. Second, our data is in agreement with what is reported in literature, showing that severe COVID-19 is associated with a cytokine storm, with our data providing novel data from the nasal mucosa. Third, our novel nasal mucosa data has significantly contributed to the identification of sub group (PCR-IgG+ SARIs) in our cohort as potentially beneficiaries of standardised COVID-19 clinical care.

5. *This manuscript would be have been more informative to focus on the comparisons b/n COVID PCR+ patients against the healthy control group.*

Response: We strongly disagree with this suggestion. We think the manuscript with the incorporated changes is highly informative and will have a significant impact on the debate on COVID-19 clinical management in low income settings. Taken together, we do not feel that reviewer three has sufficiently considered the context in which this study was conducted, nor the clinical implications of how our findings could influence care for vulnerable patients admitted to hospital in the sub Saharan African setting.

REVIEWERS' COMMENTS

Reviewer #1 (Remarks to the Author):

The authors have addressed my concerns.

Reviewer #2 (Remarks to the Author):

The authors have attempted to address the reviewers' concerns. While they have addressed most of my concerns, I still have 2 comments. Firstly, the authors provided new information that the samples were taken ~1-4 days after hospitalization. In that case, the authors should provide information whether the alterations in the cytokine analysis could be confounded by whether the patients had dexamethasone or a beta-lactam antibiotic. Secondly, with regards to the PCA shown in Figure 3a, the PCR-IgG- group did show major overlaps with the study groups as well as the control group, as both the PC1 and PC2 components would be unable to separate them apart. The claims from the authors related to Figure 3a and 3b should therefore be revised to reduce confusion. In addition, since PC2 can separate the study group from controls, the authors should provide information of the cytokines involved in PC2.

Reviewer #3 (Remarks to the Author):

as before.

Please find below our reasoned response to the remaining queries from Reviewer 2.

The authors have attempted to address the reviewers' concerns. While they have addressed most of my concerns, I still have 2 comments. Firstly, the authors provided new information that the samples were taken ~1-4 days after hospitalization. In that case, the authors should provide information whether the alterations in the cytokine analysis could be confounded by whether the patients had dexamethasone or a beta-lactam antibiotic.

Response: We have now clarified our statement within the limitations section of the discussion to specifically highlight the potential confounding caused by steroid and antibiotic use (**Discussion Section, Paragraph 6 Line 5**).

Secondly, with regards to the PCA shown in Figure 3a, the PCR-IgG- group did show major overlaps with the study groups as well as the control group, as both the PC1 and PC2 components would be unable to separate them apart. The claims from the authors related to Figure 3a and 3b should therefore be revised to reduce confusion.

Response: We have clarified this matter further in the manuscript (**Results Section, Paragraph 3 Line 7**).

In addition, since PC2 can separate the study group from controls, the authors should provide information of the cytokines involved in PC2.

Response: The cytokines that contributed to the clustering of the study groups away from the controls were IL-6, IL-4, IL-3, MIP-alpha, G-CSF, IL-1beta in nasal samples and IL-3, IL-6, Flt-3L, EGF, IFN-gamma, IL-5, IL-12p70, IP-10, IL-10, IL12p40 in serum. This information has been included in the manuscript (**Results Section, Paragraph 3 Line 7-9**).